# Spatiotemporal Heterogeneity in the Distribution of Chikungunya and Zika Virus Case Incidences during their 2014 to 2016 Epidemics in Barranquilla, Colombia

**DOI:** 10.3390/ijerph16101759

**Published:** 2019-05-18

**Authors:** Thomas C. McHale, Claudia M. Romero-Vivas, Claudio Fronterre, Pedro Arango-Padilla, Naomi R. Waterlow, Chad D. Nix, Andrew K. Falconar, Jorge Cano

**Affiliations:** 1Department of Disease Control, London School of Hygiene and Tropical Medicine, Faculty of Infectious and Tropical Diseases, London WCIE 7HT, UK; tc.mchale@gmail.com (T.C.M.); claudio.fronterre@lshtm.ac.uk (C.F.); naomi.walker1@lshtm.ac.uk (N.R.W.); chadamson91@gmail.com (C.D.N.); 2Departamento de Medicina, Universidad del Norte, Barranquilla 081007, Colombia; clromero@uninorte.edu.co (C.M.R.-V.); afalconar@uninorte.edu.co (A.K.F.); 3Programa de Prevención y Control de Enfermedades Trasmitidas por Vectores, Secretaria de Salud Distrital, Barranquilla 081007, Colombia; arangopedrojose@gmail.com

**Keywords:** Chikungunya virus, Zika virus, spatial clustering, Bayesian Poisson models, conditional autoregressive models, socioeconomic risk factors, environmental risk factors

## Abstract

Chikungunya virus (CHIKV) and Zika virus (ZIKV) have recently emerged as globally important infections. This study aimed to explore the spatiotemporal heterogeneity in the occurrence of CHIKV and ZIKV outbreaks throughout the major international seaport city of Barranquilla, Colombia in 2014 and 2016 and the potential for clustering. Incidence data were fitted using multiple Bayesian Poisson models based on multiple explanatory variables as potential risk factors identified from other studies and options for random effects. A best fit model was used to analyse their case incidence risks and identify any risk factors during their epidemics. Neighbourhoods in the northern region were hotspots for both CHIKV and ZIKV outbreaks. Additional hotspots occurred in the southwestern and some eastern/southeastern areas during their outbreaks containing part of, or immediately adjacent to, the major circular city road with its import/export cargo warehouses and harbour area. Multivariate conditional autoregressive models strongly identified higher socioeconomic strata and living in a neighbourhood near a major road as risk factors for ZIKV case incidences. These findings will help to appropriately focus vector control efforts but also challenge the belief that these infections are driven by social vulnerability and merit further study both in Barranquilla and throughout the world’s tropical and subtropical regions.

## 1. Introduction

Since 2013, the two emerging arboviruses, chikungunya virus (CHIKV) and Zika virus (ZIKV), transmitted by the same vector species (*Aedes aegypti*) as the dengue viruses (DENVs), have caused major outbreaks throughout the Americas. CHIKV is thought to have first arrived in the Caribbean region in 2013, and in 2014, it expanded to the mainland regions of Central and South America [1,2,3]. The first autochthonous ZIKV cases were reported in the northeastern region of Brazil in late 2014 [4]. By February 2016, local transmission of ZIKV was reported in over 20 countries in the Americas, including a major outbreak in Colombia, for which 65,726 cases were reported by April 2016 [5]. Both of these arboviral infections can lead to significant disability adjusted life years (DALYs) in the affected populations [6,7,8]. CHIKV infections are characterised by a high fever and debilitating symmetrical polyarthralgia/rheumatic disease in nearly all (>80%) cases, many of which result in subchronic or chronic joint disease [6,8,9,10]. By stark contrast, 80% of ZIKV infections are asymptomatic, and symptomatic patients display lower fevers and milder peripheral symptoms, such as rash, nonpurulent conjunctivitis and oedema [10,11]. However, in the Americas, ZIKV infections have been implicated in causing severe congenital abnormalities, such as foetal microcephaly, and serious autoimmune neurologic conditions in children and adults, such as Guillain–-Barré syndrome [12,13,14,15], as well as dual-gender sexual transmission [16,17].

Numerous studies have assessed the local determinants of transmission for the DENVs that in turn led to key discoveries that helped to guide public health and vector control programmes in their efforts to curb epidemics [18,19,20]. Studies have found similar transmission dynamics in CHIKV prevalence. For example, Sissoko et al. found that seroprevalence of CHIKV antibodies in Mayotte was mainly associated with lower socioeconomic status, lower education and makeshift housing [21]. Perkins et al. found that the seasonal and regional transmission patterns were similar in CHIKV compared with DENV [22]. Similarly, the spread of ZIKV through the Americas was predicted by models that accounted for the suitable environmental and socioeconomic conditions of the regions [23,24,25].

Barranquilla is a major seaport city located on the Caribbean coast and the fourth largest city in Colombia, with a population of nearly 1.4 million individuals and a population density of 7017 persons per square kilometre [26]. The city, divided into six (1–6) socioeconomic strata (SES) [5], is estimated to house 56% of its population in the poorest of those strata [27]. This city has been an epicentre of recent outbreaks of CHIKV and ZIKV [6,28]. A four-decade history of combatting DENVs has led Colombia to establish a highly functional surveillance system (SIVIGILA) for reportable DENV infections [29]. Case incidences of CHIKV and ZIKV were monitored by the same reporting system [30,31]. Previous studies have reported the strengths and limitations of this system [29,31].

In this study, we assessed the spatial and temporal heterogeneity in the CHIKV and ZIKV case incidences in the neighbourhoods of Barranquilla between 2014 and 2016. Since the neighbourhood: (a) Socioeconomic stratum [20,32,33,34,35,36], (b) population densities [35,37,38,39], (c) housing densities [33,40], (d) percentage of house or apartment dwellings [33,37,41], (e) percentage of female (domestic worker, noncommuting worker or housewife) residents [33,42,43], (f) percentage of vegetation coverage [44,45], (g) building coverage [46] versus (h) water coverage, (i) distance of residences from large water bodies [37,47], (j) parks and cemeteries [39,44,45,48] or (k) main roads [37,49] were previously identified as risk factors of *Aedes aegypti* vector breeding and human urban arboviral (DENV) disease transmissions, these parameters were evaluated as possible risk factors that influenced their reported case incidences and distributions during their epidemics in Barranquilla.

## 2. Materials and Methods

### 2.1. CHIKV and ZIKV Incidence Data

The National Health Institute and Public Health Surveillance Unit at the Health Secretary (“Instituto Nacional de Salud” and “Secretaria de Salud”) provided data on CHIKV- and ZIKV-infected cases that had been reported in Barranquilla in 2014–2016. The incidence rates were based on both clinical criteria (described below) and laboratory test confirmations. The clinical criteria for CHIKV and ZIKV infections have been regularly used along with confirmatory laboratory tests to evaluate the epidemiological characteristics of these viral infections in Latin America [6,7,35]. Cases were aggregated by neighbourhood due to the lack of more detailed geolocation information (Figure 1).

### 2.2. CHIKV- and ZIKV-Infected Case Definitions

For the Colombian public health system, CHIKV or ZIKV infections were identified based on clinically-assessed criteria with or without laboratory-test confirmations. The criteria for clinical confirmation of CHIKV infection included having a fever (>38 °C), sudden onset of severe arthralgia or arthritis and rash, not explained by other medical conditions, and living in a municipality where diagnosis of circulation by virological methods had previously occurred [50]. As such, laboratory confirmed CHIKV cases included viral isolation or an RT-PCR positive result for CHIKV cDNA, an anti-CHIKV IgM antibody positive result in an ELISA, or a four-fold increase in the IgG titres between paired (acute and convalescent phase) samples in the ELISA obtained within 2–14 days of each other, as described [50].

The criteria for clinical diagnosis of ZIKV infections included exanthema (rash) and fever (38 °C), and residence in a municipality with confirmed ZIKV transmission [51]. Additionally, ZIKV infection was considered when the patient had visited a high-risk area 15 days before the onset of symptoms, defined as zones below 2200 metres above sea level in Colombia with confirmation of local ZIKV circulation, and had one or more of the following symptoms: Pruritus (itchiness), arthralgia (joint pain), myalgia (muscle pain), headache, or malaise. Laboratory confirmation was achieved by obtaining a RT-PCR positive result for ZIKV cDNA, as described [14,50].

For both CHIKV and ZIKV, we included all clinically suspected cases, epidemiologically suspected or laboratory confirmed cases. We recorded their occurrence date as the date they were registered in the SIVIGILA system, since the date of onset of symptoms was unavailable for most patients.

### 2.3. Socioeconomic Data

The total CHIKV- and ZIKV-infected case numbers were recorded monthly by neighbourhood for the period of 2014–2016. There were 140 neighbourhoods in Barranquilla for which population data were available during this study (Appendix A).

The most recent national census for Colombia was performed in 2005, but the National Administrative Department of Statistics (DANE) had projected their populations until 2016 [26]. A Barranquilla digital map that displayed the individual neighbourhoods was obtained from the DANE website [26]. The population data and monthly incidence of CHIKV and ZIKV cases were linked within a Geographical Information System (GIS) framework to this map of neighbourhoods.

The area of each neighbourhood was used to calculate the housing and population density in square kilometres for each one of them. The percent of dwellings in each neighbourhood by category (houses vs. apartments) was obtained from the Barranquilla government website, as well as the percentage of males and females who resided in each neighbourhood [52].

In Colombia, the Central Government has defined these six socioeconomic strata to classify households based on a suite of characteristics concerning the type of construction, the number of habitable rooms, water and sanitation conditions (i.e., presence of indoor toilet, existence of piped water supply, etc.) and level of urbanisation of each home’s location (i.e., paved roads, availability of sewage system, etc.) [26]. This categorisation is mainly performed to differentially charge the different strata domiciliary public services, thereby allowing the allocation of subsidies and the collection of contributions in each area.

Each residence was assigned a socioeconomic stratum designation number from 1–6, corresponding to 1: Low–low, 2: Low, 3: Low–medium, 4: Medium, 5: Medium–high, and 6: High.

### 2.4. Environmental Data

Landsat 8 images of Barranquilla for the year 2014 (Appendix A) and 2015 (Appendix A) were obtained from the United States Geological Survey website [53]. To increase the spatial resolution of multispectral bands (provided at 30 m resolution), a pan-sharpening process was applied, excluding thermal infrared bands, using the 15 m panchromatic band. Atmospheric corrections were then conducted for bands one to eight using the DOS model-based algorithm and assuming 1% minimum reflectance [54]. Radiance and ground reflectance were subsequently obtained for each band [55]. Band combinations for ideal enhancement of vegetation were obtained from openweathermap.org [56]. A traditional infrared colour map with the band combinations of 5-4-3 was used to highlight areas of vegetation. A modification of the normalised difference water index (MNDWI) image was then created to highlight areas of water [57]. These raster images were combined to define areas of vegetation, building and water coverage. Finally, zonal histogram statistics were extracted from the raster to calculate the percentage surface area of each neighbourhood that was occupied by vegetation, buildings or water. Next, the MNDWI image was used to calculate the average distance of each neighbourhood from large bodies of water identified in the raster. The Euclidian (straight-line) distance tool in ArcGIS was used to calculate the straight-line distance from the centroid of each neighbourhood to the nearest pixel of water identified in the MNDWI. All geographical processing was performed in ArcGIS 10.3 (ESRI Inc., Redlands, CA, USA) [58].

A map layer in shapefile format that identified parks, forests and cemeteries throughout the city was downloaded from the bbike.org website [59]. We used a high-resolution satellite map from the city to verify the accuracy of the delineated green areas. The Euclidian distance of these identified parks, forests or cemeteries throughout the city was then calculated using the same approach employed to estimate distance of the residents’ dwellings from water bodies. We then referred to this variable as ‘distance to parks or cemeteries.’ Each of the covariates listed above were then linked to the map of neighbourhoods in Barranquilla along with the monthly incidence of CHIKV and ZIKV cases. Previous studies have shown that vegetation coverage and distance to parks [48] as well as distance to large water bodies [37,47] have been important predictors of *Aedes aegypti*-borne infections. Thus, we included them in our multivariate regression models.

### 2.5. Statistical Analysis

#### 2.5.1. CHIKV and ZIKV Incidence Rate

The total incidence of surveillance-defined or laboratory-confirmed CHIKV and ZIKV cases over the study period was linked to the neighbourhood shapefile. The number of recorded cases of each of these viral infections was divided by the population of each neighbourhood and multiplied by 10,000 to achieve an incidence rate of cases per neighbourhood per 10,000 persons at risk. The incidence was calculated for CHIKV and ZIKV for each individual monthly. The spatial structure of CHIKV and ZIKV case incidences over the study period of 2014 to 2016 was mapped using ArcGIS 10.3. (ESRI Inc., Redlands CA, USA) [58].

#### 2.5.2. Analysis of Spatial Clustering

Exploratory analysis of spatial clustering was conducted using a global Moran’s I statistic and local indicators of spatial association (LISA). The univariate Moran’s I statistic was used to explore the existence of overall spatial autocorrelation for the total incidence of CHIKV and ZIKV cases throughout Barranquilla, whereas LISA analysis was implemented to explore local spatial autocorrelation and identify hotspots among neighbourhoods. An area of high–high correlation (hotspot) indicated that the estimated incidence exceeded the neighbourhood average. An area of high–low correlation indicates that the central neighbourhood was negatively correlated with its neighbours, having a higher than average incidence, compared to a lower than average incidence in the neighbours. Alternatively, low–low correlation indicated that neighbourhoods were correlated together with lower than average incidence. However, low–high indicated negative correlation with the central neighbourhood having lower than average incidence compared to a higher than average incidence in its neighbours. More details about the implementation of these statistics are provided in Appendix A.

Differential Moran’s I statistic and LISA analyses were used to explore the variation of spatial autocorrelation over time. In these differential analyses, a hotspot indicated that a higher than average case incidence in the highlighted neighbourhood during the first time period led to a higher than average case incidence in the bordering neighbourhoods during the following time period. These analyses were performed for the CHIKV and ZIKV cases for each month of their epidemics: July 2014 to April 2015 for CHIKV and October 2015 to June 2016 for ZIKV cases (Appendix A). GeoDa software was used to perform the exploratory spatial clustering analyses [60,61,62,63].

#### 2.5.3. Missing-Data Imputation 

The original dataset contained missing observations for some of the explanatory variables: Socioeconomic stratum, housing density and the percentages of females, house dwellings and percentage of vegetation coverage. These missing values were therefore estimated using a Besag–York–Mollié conditional autoregressive (CAR) spatial model [64,65]. Details on the Bayesian model implementation are provided in a Appendix A. For each variable, an intercept-only model was fitted, and the missing observations were imputed using the resultant predictive posterior median.

#### 2.5.4. Bayesian Statistical Modelling

Since the dataset was too sparse and noisy to fit a spatiotemporal model, the monthly counts of CHIKV- and ZIKV-infected cases were aggregated to consider only the potential spatial variation of the infection over the entire study period. A standardised incidence rate (SIR) for each neighbourhood was calculated taking the distribution of cases and population into account, as follows:
$$SIR_{i}=\frac{O_{i}}{E_{i}}$$ where $O_{i}$ is the observed number of cases, $E_{i}=rP_{i}$ is the expected number of cases, $P_{i}$ is the population and r=
$\frac{\sum_{i=1}^{n}O_{i}}{\sum_{i=1}^{n}P_{i}}$ is the overall incidence ratio.

This metric was used to estimate the risk of infected case incidence associated with residence in each neighbourhood.

SIR values across neighbourhoods were fitted using multiple Bayesian Poisson models with a range of explanatory variables as potential risk factors and different options for random effects with: (i) No random effects, (ii) independent random effects and (iii) spatially correlated random effects, implemented through a CAR.

All explanatory covariates were mean centred. In this way the intercept was interpretable as the average global SIR, and the SES regression coefficients were calculated in reference to the lowest SES level. The adequacy of these models was explored using standard posterior predictive checks and a final comparison, to select the model that best fit the data, was performed using the deviance information criterion (DIC) [66]. Details on the specifications of these Bayesian Poisson models are given in a Appendix A.

## 3. Results

### 3.1. Description of Socioeconomic and Environmental Data

Table 1 shows the total incidence of CHIKV and ZIKV during the 2014–2016 period and a summary of descriptive statistics for the socioeconomic and environmental factors analysed in this study. Neighbourhoods of the lowest socioeconomic strata (SES 1) were located in the southwestern and eastern areas, while those of the highest SES (SES 5 and 6) were located in the northern areas of the city (Figure 2B). These neighbourhoods with the lowest SES (SES 1) in the southwestern areas of the city had the highest percentage of houses, while many SES 6 neighbourhoods in these northern areas had the lowest percentage of houses (Figure 2A).

For 21.3% (*n* = 30) of the neighbourhoods, there were no data of their housing densities, nor were there data on the percentage of houses versus apartments for 13.5% (*n* = 19) of the neighbourhoods. Missing data were not, however, a concern for all the other variables, since those data were not available for fewer than 5% of the neighbourhoods. Amongst the recorded cases in the original dataset, the resident neighbourhoods were unknown for 5.9% (*n* = 110) and 10.5% (*n* = 746) of the patients with CHIKV or ZIKV infections, respectively.

### 3.2. Analysis of Chikungunya Virus Outbreaks

During the study period, there were a total of 1865 cases of CHIKV in Barranquilla. The total incidence of CHIKV during the study period was 16.4 per 10,000 persons, and the incidence was 12.1, 3.65 and 0.596 per 10,000 persons in 2014, 2015 and 2016, respectively. There was a single distinct peak of CHIKV cases that began in the 4th quarter of 2014, peaked in November to December and continued until the end of the 1st quarter of 2015 (Appendix A).

When the CHIKV incidence rate was displayed in maps by neighbourhoods (Appendix A), there were two clear areas of high incidence rates (within the 5th quantile), particularly in the northern area and the southwestern area, located next to the main western circular road route around the city. The global univariate Moran’s I statistic for the total incidence of CHIKV-infected cases during the study period was 0.091 (pseudo *p*-value 0.0026), suggesting a global clustering pattern across the study area. The LISA cluster map showed one significant cluster of high–high incidence (‘hotspot’) in the southwestern (neighbourhoods: 1204, 1205, 1206, and, 3704) area of the city (Appendix A for neighbourhood codes).

The analysis of local clustering over time (differential LISA analysis) showed a clear evolution of the CHIKV epidemic during the months of peak incidence from July 2014 to January 2015. Hotspots appeared in the northwestern area of the city at outbreak onset (neighbourhoods: 0301, 0305, 0307 and 0311 and later in the epidemic also together with 0102, 0201, and 0312), which then moved to the southwestern area (neighbourhoods: 1204, 1205, 1208, 3704). The hotspot area expanded from a single neighbourhood (1206: El Pueblo), located immediately adjacent to the main warehouses used for delivery and export of cargo to and from the city by road and sea, from August to September to three neighbourhoods (1204, 1205, and 3704) in October to November (Figure 3). Importantly, the main circular road in the southwest of the city either passed directly through (1204) or immediately adjacent to (1205, 1206 and 3704) them (Figure 1 and Figure 3 and Appendix A for neighbourhood names). Subsequently, hotspots appeared to move to the margins of the city in the northern and southeastern regions, where the incidence rate increased sharply (Figure 3).

The Bayesian Poisson model of CHIKV SIR values fitted with the explanatory variables (SES, population density, housing density, percentage house dwellings, female resident ratio, vegetation coverage, distance to major roads, distance to parks or cemeteries, and distance to water bodies) and with no random effects showed a poor fitting performance of the incidence data (DIC: 1718.3) (Table 2). Looking at the posterior predictive distributions and the posterior predictive *p*-value, there were clear signs of over-dispersion, since the high degree of variability in the observed SIR values was not accounted for by the model. We therefore introduced a set of independent random effects using the model 2 approach (Text S2) to capture the remaining variability. Since the model 1 approach showed a spatial structure on the distribution of CHIKV SIR values across Barranquilla, we assessed the persisting spatial correlation on the residuals for the fitted model 2, controlling for spatially varying explanatory variables. Indeed, we still observed significant global residual dependence (Moran’s I statistic: 0.0716, *p*-value: 0.04); thus, we opted to employ a set of spatially structured random effects that could cope with a global spatial correlation (model 3 approach). Model 3, the globally smooth CAR model with spatially correlated random effects (Text S2) showed the better fitting performance, as indicated by its DIC value (DIC: 733.4) (Table 2), and the resulting residuals no longer showed any presence of spatial dependency (Moran’s I statistic: 0.0202, *p*-value: 0.08), thereby reinforcing our model selection. Figure 4 shows the crude and fitted CHIKV SIR values throughout Barranquilla.

Table 3 shows the regression coefficients for the spatial fixed effects considered to fit CHIKV SIR values, based on the model 3 approach. Risk of CHIKV case incidences appeared to slightly decrease with increasing percentage of vegetation coverage (Coefficient 0.99, 95% CI 0.97, 1.00). The poorest neighbourhoods, based on three levels of SES (quantiles), were the most at risk for CHIKV infections (reference low SES, medium SES coefficient 0.40, 95% CI 0.22, 0.72; high SES 0.61, 95% CI 0.24, 1.52). In our model, population density, housing density, and percent of dwellings that were houses showed little effect on the incidence of CHIKV-infected cases, while infection risk was increased by 10% for a unit percentage increase of female residents. None of these associations were, however, statistically significant.

### 3.3. Analysis of Zika Virus Outbreaks

During the study period, there were a total of 7029 ZIKV-infected cases in Barranquilla. The total incidence of ZIKV was 61.7 cases per 10,000 persons at risk. The incidence of ZIKV was 0, 12.3 and 49.4 cases per 10,000 persons in 2014, 2015 and 2016, respectively. There was a single distinct peak of ZIKV cases that began at the beginning of the 4th quarter of 2015 and continued until the end of the 1st quarter of 2016 (Appendix A).

The map that displayed the total ZIKV incidence rate by neighbourhood showed a focus of high incidence in parts of the northern and the southwestern areas of the city (Appendix A). The global univariate Moran’s I statistic for the total incidence of ZIKV during the study period was 0.031 (pseudo *p*-value 0.191); thus, there was not a clear global clustering pattern for this infection in Barranquilla. However, the LISA analysis showed three distinct small hotspots in the northern, southwestern and southern areas of the city (Appendix A).

The clustering of cases during the ZIKV outbreak was less dramatic compared to that of the CHIKV epidemic, as revealed by the differential LISA analysis. However, a clear focus of hotspots occurred at the beginning of the outbreak in the northwestern (neighbourhoods: 0305, 0307, 0311 and 0312) area of the city, where the 0307, 0311, and 0312 neighbourhoods were located on a major road crossing the city from the main circular road (Figure 1 and Figure 5). These hotspots then evolved to the northeastern area during November to December 2015 and then moved to the margins of the eastern side of the city (neighbourhoods: 2101, 2102, 0803 and 0804), located next to the main circular road around the city and also neighbourhood 2103, located around the main international city harbour on the Magdalena River from January to February 2016. As the outbreak waned from February to March, the hotspots returned to the northeastern area of the city (Figure 5).

As with the CHIKV SIR values, the Bayesian Poisson model of the ZIKV SIR values fitted with the explanatory variables and with no random effects showed a poor fitting performance of the incidence data (DIC: 2815.7) (Table 2). Thus, we introduced a set of independent random effects using the model 2 approach (Text S2) to capture the remaining variability. Unlike the CHIKV SIR model 2, we did not find any significant residual spatial correlation at a global scale (Moran’s I statistic: 0.0101, *p*-value: 0.329) upon fitting the ZIKV SIR value data based on the model 2 approach. However, a local Moran I statistic clearly showed the presence of some residual local spatial dependence (Appendix A). Therefore, we fitted a locally smooth CAR model with spatially correlated random effects (Text S2) that demonstrated a better-fitting performance compared to previous modelling approaches (DIC: 967.6) (Table 2). Figure 6 shows the crude and fitted ZIKV SIR values throughout Barranquilla.

Table 4 displays the regression coefficients for the spatial fixed effects considered to fit the ZIKV SIR values based on the model 4 approach. The risk for ZIKV case incidences did not vary or varied very little according to the distance of dwellings from parks or cemeteries and water bodies, or the percentage of vegetation coverage. Surprisingly, the proportion of female residents also did not appear to be associated with the ZIKV case incidence. However, the richest neighbourhoods (SES 5 and 6) were significantly more at risk for ZIKV infections compared to those of the low and medium SES (*p*-value < 0.05) (Table 4 and Figure 2B). Living in a neighbourhood that was a greater distance from major roads also significantly decreased the risk of ZIKV case incidence (Table 4 and Figure 1).

## 4. Discussion

The global cluster analysis of the CHIKV-infected case incidences identified hotspots of transmission in the southwestern area of the city. The monthly analysis of local autocorrelation showed that the CHIKV epidemic began within hotspots in the northern area and then moved quickly to the southwestern region of this city, affecting mostly the poorest communities, following the main western circular route (circumvent roads) that was also connected via a road with the principal truck transport depot and the southern road route to the city of Cartagena (Figure 1 and Figure 3). The epidemic then appeared to spread back to the north of the city and to the southeast of the city via the main circular road. However, it may have also been reintroduced via boats arriving in the international harbour area on the Magdalena River (neighbourhood 2103) or via the road route (southeast direction) to the city of Santa Marta, and finally being transported from those northern neighbourhoods to the adjacent ones in December to January 2015. While there was little global spatial autocorrelation for the ZIKV-infected case incidence, the monthly analysis of local autocorrelation showed a high-risk area in the northern neighbourhoods of the city as the outbreak began. As the outbreak intensified, the central–eastern neighbourhoods of the city became important hotspots. It seemed that the ZIKV epidemic spread to the eastern part of the city probably through the main circular road, and a major road that crossed from the northwestern to the mideastern part of the city, in a clockwise direction (Figure 1 and Figure 5).

Using Bayesian Poisson regression allowing for random effects and controlling for all the covariates (fixed effects) in Table 1, we found weak evidence that lower SES was associated with higher SIR of CHIKV infection. We, however, found very strong evidence that higher SES and living nearer to main roads had a higher risk for ZIKV infection.

We believe this is the first report that provides strong evidence that living in higher SES is a risk factor for ZIKV case incidences. Furthermore, the northern neighbourhoods of Barranquilla are of these higher SES, and these neighbourhoods located there were the first to report high infection incidences and appeared to drive both the CHIKV and ZIKV outbreaks. Unfortunately, the local health authority’s vector surveillance and control teams stated that it has been difficult to implement their work in areas of higher SES due to them being walled and gated communities (‘conjuntos’) with private security guards providing restricted access. Thus, limited, or lack of, access to identify and treat vector-breeding sites in these areas may have accounted for their higher ZIKV case incidence. 

Alternatively, these findings may indicate a reporting bias resulting from these residents having better access to health facilities with resultant higher positive diagnosis rates. The issue of reporting bias was likely exacerbated during the ZIKV epidemic by the widespread media coverage drawn by the association of ZIKV with congenital defects, sexual transmission and severe autoimmune diseases [14,15,16,17]. Those with easier access to clinics would have been more likely to present to clinic with milder symptoms and receive the diagnosis of ZIKV as it was sweeping through the city. In addition, people living in higher socioeconomic strata would be more likely to be receiving regular prenatal healthcare, where there is already an over diagnosis of ZIKV [67]. This is in stark contrast to the presentation of CHIKV infection, which results in very high fever and debilitating arthralgias. Our findings may therefore be supported by finding that poverty was associated with reduced ZIKV case reporting in another recent study [49]. Thus, the health authority team’s access to these high-risk areas is urgently required to assess whether these results were due to the lack of vector surveillance and control or a higher incidence of case reporting. To our knowledge, higher incidences of vector-borne diseases have usually been associated with lower socioeconomic status [32,33,35,41], often due to poor sanitation and limited access to piped water leading to their need to store domestic water supplies in large water containers, which act as principal breeding sites for the *Aedes aegypti* vector species, as reported in our study sites [68,69,70]. The two groups at most risk for DENV infections in Itaipu, Brazil were the lowest-income population and a group in a slightly higher SES who had the funds to buy and store domestic water in large tanks [71]. The evidence we present here indicates that either the CHIKV and ZIKV epidemics occurred in different ecological niches or the case reporting of ZIKV infection was novel amongst the arboviral infections.

Through conversations with the vector control teams, the most common breeding sites in apartments were flowerpots. While these were commonly positive for *Aedes aegypti* pupae, they do not produce large numbers of pupae and therefore resultant adult vectors [70,72]. Houses, on the other hand, are much more likely to have gardens or yards with or without patios that contain used or discarded water-holding containers, as well as very large domestic water storage containers (drums and tanks) that are the principal *Aedes aegypti* breeding sites, holding 72% to 78% and 65% to 96% of the pupae populations in the wet and dry seasons, respectively, in three study neighbourhoods in Barranquilla [70]. Previous studies have found that residents living in apartments were at a reduced risk for DENV infections [33,41,73], probably due to reduced availability of suitable sites for *Aedes aegypti* breeding [41,74]. Our data supported these findings since the southwestern area of the city, where the neighbourhoods contained much higher percentages of houses (Figure 2A), was an important hotspot during the CHIKV outbreak (Figure 3). Much lower combined *Aedes* spp. premise indexes, an indicator of premises with active breeding sites, were reported in apartments rather than compound houses in Singapore [41]. Surprisingly, however, *Aedes aegypti* was shown to have adapted to breeding in the ‘ecosystems’ present on each floor of high-rise apartment buildings in Kuala Lumpur, Malaysia [75].

Drawing spatial distinctions between the CHIKV and ZIKV cases assumed that the infected cases reported for these analyses were accurately identified. However, there are known limitations for accurate clinical discrimination amongst arboviral infections, especially since financial restraints made it impossible to perform laboratory confirmation on all suspected cases. Previous projects in Barranquilla further evaluated the frequency with which these methods were used and found that only 1% of ZIKV and 1.7% of CHIKV-infected cases were laboratory confirmed [30,31]. Since the CHIKV and ZIKV outbreaks occurred in separate time periods, any possible misclassifications were more likely to have been due to the DENVs, of which all four serotypes were previously shown to be endemic in Barranquilla [76,77]. While some studies have been limited by the cross-reactivity of IgM and IgG antibodies generated by patients with DENV and ZIKV (both flaviviruses) infections [78,79], all the laboratory-confirmed ZIKV cases in our study were identified by a positive ZIKV-specific RT-PCR.

We minimised the limitations of the modifiable area unit problem (MAUP) [80,81,82] in our study by using neighbourhoods as the unit of analysis that was the smallest possible resolution available to us. However, while these analyses identified neighbourhood-level risk factors, the ecological fallacy prevented us from interpolating this to individual risk level. The statistical regression was limited by the possibility of residual confounding. There were several variables that we were unable to evaluate. The ethnicity of the people living in each neighbourhood may be important, as shown in other studies, where Afro-Colombians were shown to have lower DENV-infected case incidences [35,83]. Occupation, especially since it affects the movement of populations towards and away from neighbourhoods with various disease risk factors, may play an important role, as was reported for DENV-infected case incidences in poor neighbourhoods with high populations of young (<15 years old) [33,84], illiterate and unemployed people in Cali, Colombia [42]. Housing data were unavailable for 30% of the neighbourhoods, while the percentage of dwellings that were houses or apartments was unavailable for 19% of the neighbourhoods. We used a multiple imputation approach to mitigate the effect of unavailable data. This assumed that the unavailable data were missing at random and the imputation model was correctly specified [85].

The findings in our study are novel in their attempt to correlate previous epidemiological patterns of DENV-infected case incidence neighbourhood risk factors discussed above with those of recent outbreaks of CHIKV and ZIKV. Our spatial exploration clearly identified neighbourhoods that had been hotspots responsible for driving the epidemic waves of CHIKV and ZIKV. It is therefore essential that these hotspots be constantly targeted by the local health authority to reduce *Aedes aegypti* breeding sites to a minimum and quickly act to target all areas in and surrounding them when any cases of CHIV or ZIKV are reported. The risk factor analysis provided strong evidence that higher socioeconomic status and living in a residence closer to the main roads were risk factors for ZIKV case incidence. In Barranquilla, the Public Health Department should be empowered to commit more resources to these areas of the city. While it may seem counterintuitive to direct public resources to areas of higher SES, it is urgently required to determine whether higher *Aedes aegypti* vector populations or increased residents’ reporting accounts for our findings. If the former case was responsible, then the public health authorities must identify and destroy the vector breeding sites through drainage or larvicidal treatment.

Further research also needs to be performed to account for differences in vector breeding, dispersal and disease transmission. During the CHIKV epidemic, the possible introduction of the virus into (a) the northern neighbourhoods and (b) the western neighbourhood (1206), located immediately adjacent to the import/export cargo warehouses via the main circular roads and further dispersal by the main road, as well as its possible later importation into (c) neighbourhood 2103 via cargo ships docking at the major international harbour, is consistent with both major roads [37,86,87] and seaports/rivers [87,88,89] being important routes for the importation, movement and dispersal of arboviral-infected *Aedes aegypti* and humans.

More in-depth data about housing vulnerability could include type of building material, exposure to the outside, and presence of air conditioning, since home air conditioning was a major factor accounting for the much lower DENV sero-positivity for the residents in a study site in Texas, USA compared to that in a twined town in Mexico [90]. Collecting this information could reduce the residual bias associated with our crude measure of housing type. Further analysis should assess these arboviral case incidences with the residents’ occupation and other forms of population movement. More detailed vector surveillance is also needed throughout the city to elucidate why the risk factors identified in this study are associated with higher transmission. Although adult *Aedes aegypti* are known to fly up to 2 km if required, they usually fly much shorter distances (e.g., 10–150 m) to find blood meals and breeding sites [91], and there were no natural barriers (e.g., large lakes, parks, hills or valleys) in this urban study site since we did not study any of the (only poor non-urban) neighbourhoods located on the opposite (eastern) bank of the Magdalena River. As no effective treatments or vaccines exist for CHIKV and ZIKV, prevention is still the primary method of control [8,12,92].

## 5. Conclusions

This study provided an evidence-based framework that public health programmes can use to efficiently target the drivers of future epidemics of CHIKV and ZIKV infections. Our spatial analysis identified key areas of Barranquilla that acted as hotspots during the outbreaks of CHIKV and ZIKV. Furthermore, we clearly showed that living in the high SES was a risk factor for ZIKV case incidence, though this may have been driven by biased case reporting. These richer areas were important hotpots of case incidence during both outbreaks. This novel finding challenges the logic that *Ae. aegypti*-borne infections are driven by social vulnerability and merits further study both in Barranquilla and throughout the tropical and subtropical areas of the world.

The findings in this article have been relayed to the local health authority (*Distrisalud*) so that their surveillance and vector control teams can more effectively focus their efforts in the neighbourhoods which contain part of or lie immediately adjacent to these major city roads, particularly those also located in: (a) The western region of the city near the main international cargo storage warehouses, (b) the international harbour area and (c) the high SES areas within these northern areas of the city.

## Figures and Tables

**Figure 1 ijerph-16-01759-f001:**
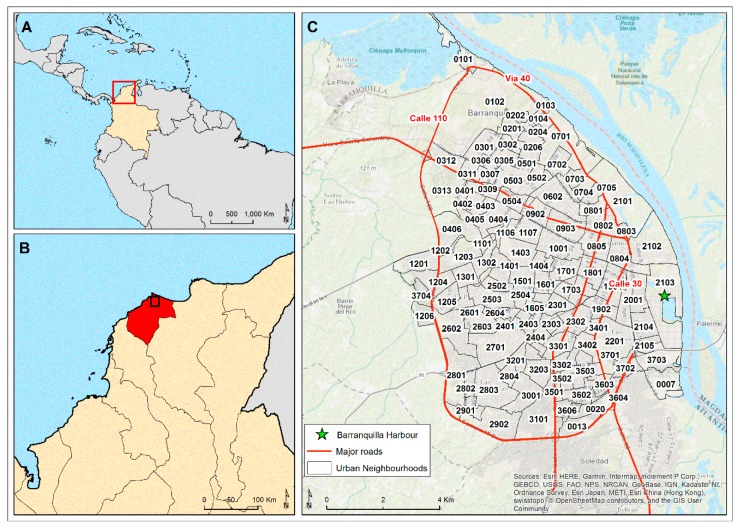
Map of the Barranquilla, Colombia study site. The neighbourhood names are listed in Appendix A.

**Figure 2 ijerph-16-01759-f002:**
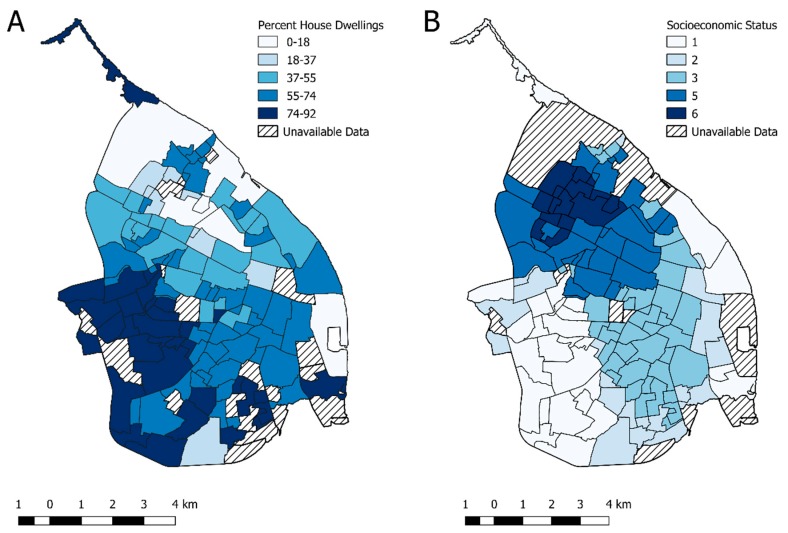
Topographical distribution of (**A**) the percent of dwellings that were houses and (**B**) the socioeconomic strata designation for each neighbourhood.

**Figure 3 ijerph-16-01759-f003:**
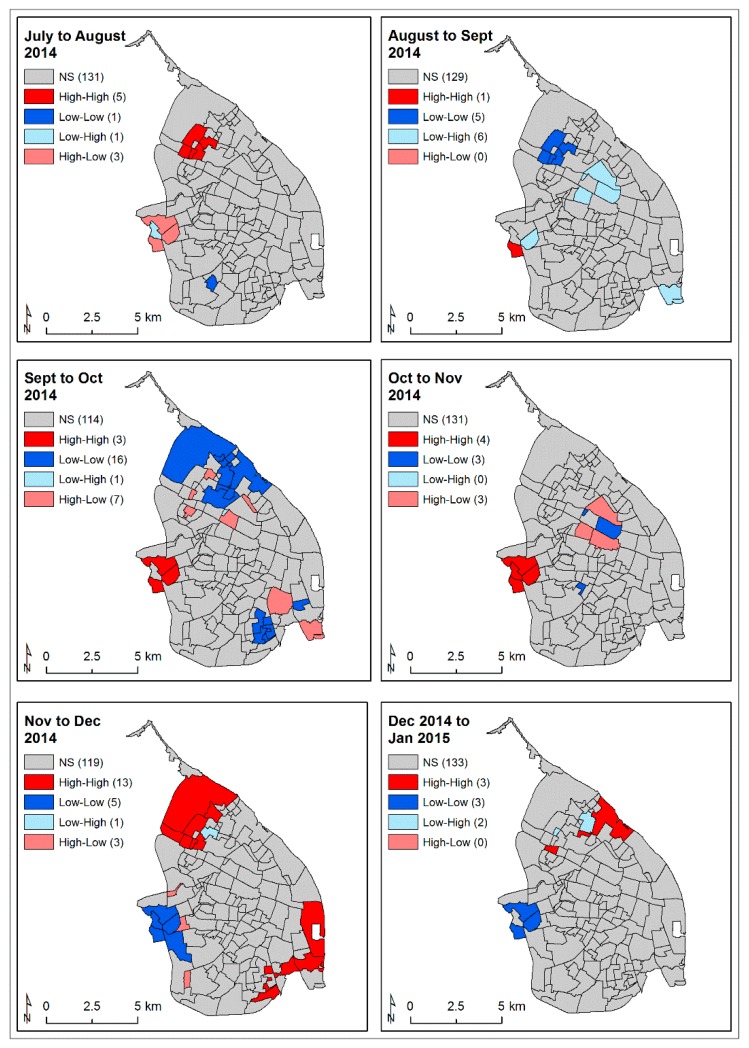
Differential monthly LISA clusters of CHIKV-infected cases during the period of peak incidence (July 2014–January 2015). High–high to low–low colour-coded neighbourhoods (see Materials and Methods 2.5.2) were significant, defined as *p* < 0.05. NS: Not significant (grey colour).

**Figure 4 ijerph-16-01759-f004:**
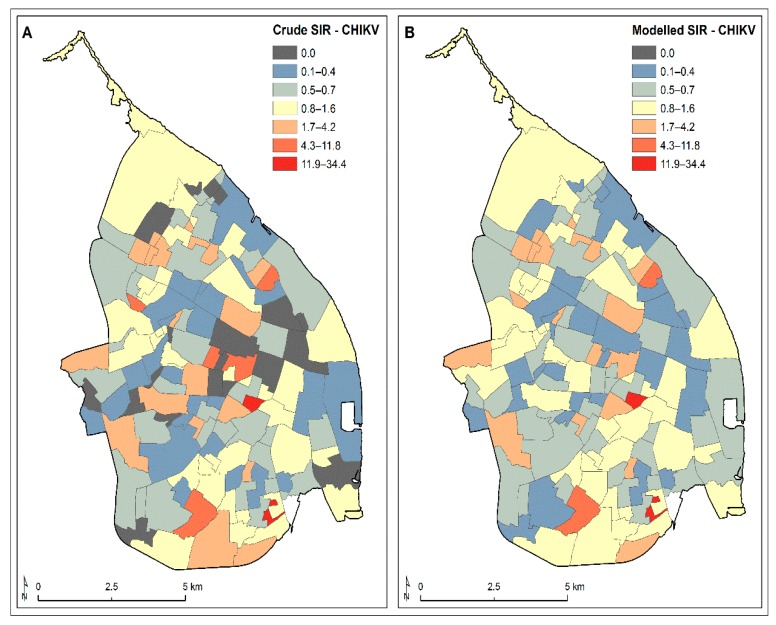
Crude and modelled CHIKV standardised incidence rate (SIR) values by neighbourhood during its 2014–2015 epidemic in Barranquilla. (**A**) Fitted globally smooth CAR model for the SIR with spatially correlated random effects and (**B**) the modelled SIR with spatially correlated random effects.

**Figure 5 ijerph-16-01759-f005:**
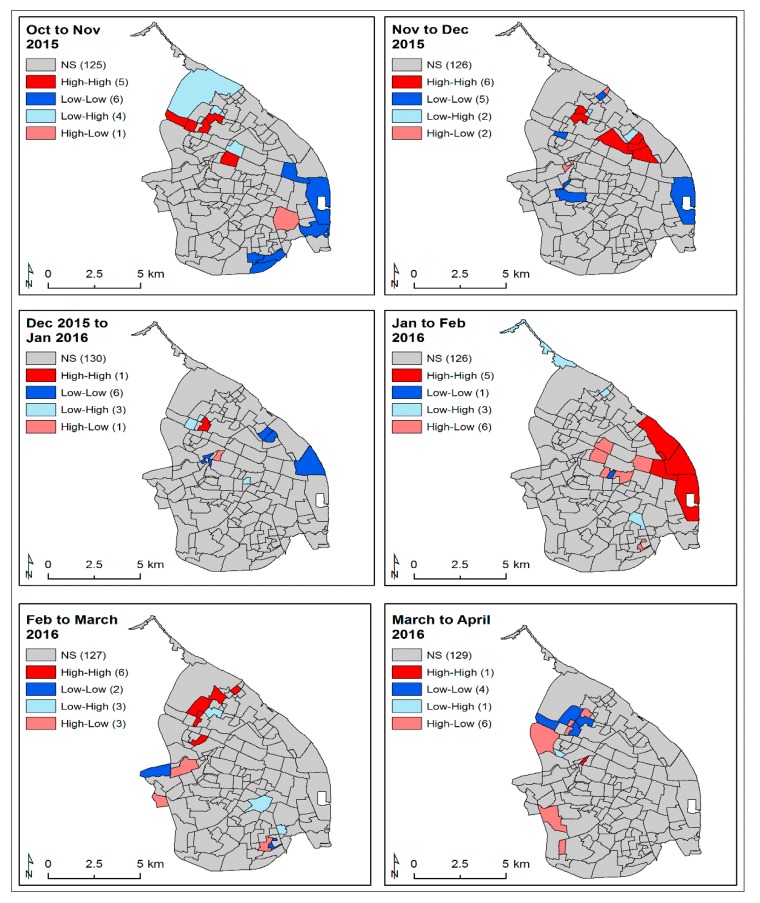
Differential monthly LISA clusters of ZIKV cases during the period of peak incidence (October 2015–April 2016). High–high to low–low colour-coded neighbourhoods (see Materials and Methods 2.5.2) were significant, defined as *p* < 0.05. NS: Not significant (grey colour).

**Figure 6 ijerph-16-01759-f006:**
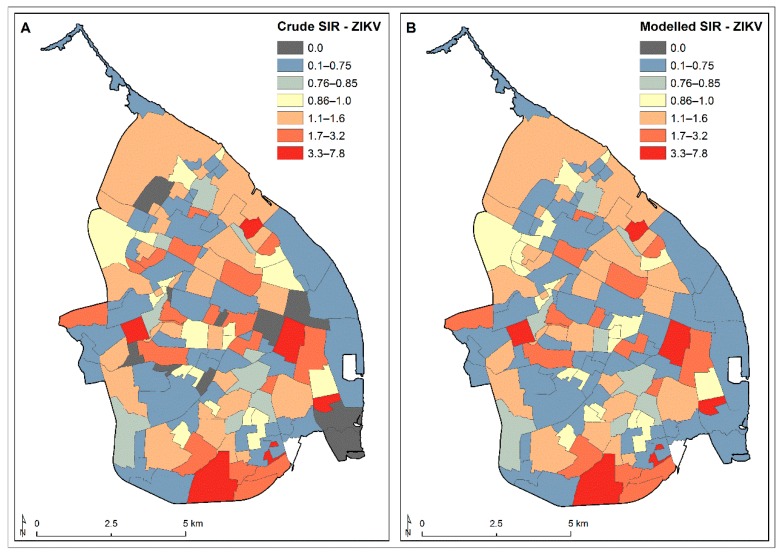
Crude and modelled ZIKV standardised incidence rate (SIR) values by neighbourhood during its 2015–2016 epidemic in Barranquilla. (**A**) Fitted locally smooth CAR model for SIR with spatially correlated random effects and (**B**) the modelled SIR with spatially correlated random effects.

**Table 1 ijerph-16-01759-t001:** The descriptive statistics of the neighbourhoods (*n* = 140) for Chikungunya virus (CHIKV)- and Zika virus (ZIKV)-infected cases and risk factors that were included in the regression analyses. NA = not available.

	Median	Percentile		NA, % (*n*)
		5th	95th	
Incidence of CHIKV per 10,000 residents	10.6	0	48.4	0.00 (0)
Incidence of ZIKV per 10,000 residents	50.3	0	157	0.00 (0)
Socioeconomic stratum	3	1	6	4.96 (7)
Population density (persons/km^2^)	16.6	1.96	41.4	0.00 (0)
Housing density (dwellings/km^2^)	3.50	0.776	6.44	21.3 (30)
Percent house dwellings	65.8	23.6	85.2	13.5 (19)
Percent apartment dwellings	33.7	11.0	71.2	13.5 (19)
Percent male	46.9	41.1	50.8	3.55 (5)
Percent female	53.1	49.2	58.9	3.55 (5)
Percent vegetation coverage	4.88	0.248	53.4	1.42 (2)
Percent building coverage	95.1	46.7	99.8	1.42 (2)
Percent water coverage	0	0	0.0792	1.42 (2)
Distance from major roads to neighbourhood centroids (metres)	639.9	265.6	1164.5	0.00 (0)
Distance from large water bodies (metres)	2096	334	3961	0.71 (1)
Distance from parks or cemeteries (metres)	473	146	2047	0.71 (1)

**Table 2 ijerph-16-01759-t002:** Deviance information criterion (DIC) values for the Bayesian Poisson models. See Methods Section 2.5.4 for model details.

Type of Fitted Models	CHIKV	ZIKV
Model 1: No random effects	1718.3	2815.7
Model 2: Independent random effects	733.5	974.9
Model 3: Globally smooth CAR	733.4	
Model 4: Locally smooth CAR		967.6

**Table 3 ijerph-16-01759-t003:** The posterior median and 95% credible intervals for the fixed effects of the final model for the CHIKV standardised incidence rate (SIR) values in 2014–2016. These parameters are reported on the exponential scale so that the effects could be interpreted as multiplicative on the SIR. Socioeconomic strata (SES) were grouped in three categories: High (SES 5 and 6), medium (SES 3 and 4) and low (SES 1 and 2), and the regression coefficients were obtained for the medium SES and high SES compared to the low SES.

	Regression Coefficient	95% CI
Intercept	1.09	(0.71, 1.68)
Medium SES (ref. class low)	0.40	(0.22, 0.72)
High SES (ref. class low)	0.61	(0.24, 1.52)
Population density	0.99	(0.97, 1.02)
Housing density	0.98	(0.86, 1.09)
Percent house dwellings	1.01	(1.00, 1.03)
Percent female	1.10	(0.99, 1.24)
Percent vegetation coverage	0.99	(0.97, 1.00)
Distance from major roads	1.08	(0.77, 1.50)
Distance from large water bodies	1.08	(0.91, 1.29)
Distance from parks or cemeteries	1.15	(0.81, 1.65)

**Table 4 ijerph-16-01759-t004:** The posterior median and 95% credible intervals for the fixed effects of the final model for the ZIKV standardised incidence rate (SIR) values in 2014–2015. These parameters were reported on the exponential scale so that the effect could be interpreted as multiplicative on the SIR values. Socioeconomic strata (SES) were grouped in three categories: High (SES 5 and 6), medium (SES 3 and 4) and low (SES 1 and 2), and the regression coefficients were obtained for the medium SES and high SES compared to the low SES.

	Regression Coefficient	95% CI
Intercept	0.79	(0.64, 0.96)
Medium SES (ref. class low)	0.83	(0.62, 1.09)
High SES (ref. class low) *	2.00	(1.23, 3.09)
Population density	1.01	(1.00, 1.02)
Housing density	0.94	(0.89, 0.99)
Percent house dwellings	1.01	(1.00, 1.02)
Percent female	1.05	(0.99, 1.10)
Percent vegetation coverage	1.00	(0.99, 1.00)
Distance from major roads *	0.78	(0.66, 0.91)
Distance from large water bodies	1.15	(1.04, 1.26)
Distance from parks or cemeteries	1.06	(0.90, 1.25)

* Significant association (*p*-value < 0.05).

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
