# Peer review of "Spatiotemporal Heterogeneity in the Distribution of Chikungunya and Zika Virus Case Incidences during their 2014 to 2016 Epidemics in Barranquilla, Colombia"

_ijerph, 2019, doi:10.3390/ijerph16101759_

Round 1
Reviewer 1 Report
see also attached file. I added my recent comments in RED after my Original comment from February
As a medical entomologist working in Public health institution on Aedes vector in Caribbean and Europe, I read the manuscript by Mc Hale et al. submitted to IJERPH with great interest. The paper tells a compelling story, challenging the common hypotheses on that Aedes borne infections are driven by social vulnerability. I highly appreciate the daunting task of the authors to bring all the data together, analyse and model it and draw conclusions from it. However, I have some issues that I would like to raise and I hope the authors will be keen to answer them
Scientific questions
Unfortunately the authors avoided to address the most curious result of their study “why the spatial clustering for CHIKV cases differs from those for ZIKV”. They mention it in page 15, line 390-391 but provide no explanation or hypothesis for this finding; it is not even discussed. While reading the paper I was eager to learn what the authors’ take on the issue was, but it was not addressed. The authors treat the two outbreaks completely independent while they both are transmitted by the same vector. Would there be a spatial difference in inhabitants between clinical outcome after infection between neighborhoods. Would there be a reason to assume that people with higher SES would report Zika but not chikungunya. Would there be a reason that areas with SES where more difficult to access during the Zika outbreak then the chikungunya outbreak. I largely miss the discussion on this issue. Only partially addressed. The authors added info p22 lines481-483, and lines493-494, p23 line 562 (the best one). The authors admit to test correlations with potential risk factors with the incidence data but still no insights whether it is driven by mosquito density or reporting bias. Still puzzling also for me how the strong difference between CHIKV and ZIKV occurred but I am happy they addressed here but the authors “forgot” to implement it in the conclusion (therefore please omit “we clearly showed that living in the high SES was a risk factor for ZIKV transmission”; you CAN state that they are important hotpots of case incidence, acknowledging that the high incidence is either real or biased. The fact that the latter is not implemented in the text in the conclusion I get the impression that the authors are more data crunchers then public health professionals who wants to know what is the factor /cause behind the data so that interventions can be applied correctly. It is still weird that the incidence of diseases that are transmitted by the same vector have different environmental drivers. I wished the authors would be a bit more impressed with this difference. A statistical comparison between the CHIKV and ZIKV hotspots would be an interesting exercise.
Although the results indicated the differences I am not satisfied with the rational behind it and I am uncomfortable with the authors statement in the first sentence of the conclusion (p16, line 469-470) that the study provide evidence based framework that the PH programmes can use to effectively target drivers of future epidemic. As a public health expert I still wonder how I could act based on the findings of the paper. Partially addressed. I am wondering whether not just random factors determine the introduction of a virus in a neighborhood and that the vector density in all neigborhoods is above threshold for efficient transmission and for that reason other possible random factors play a more important role. The authors found a pattern in two outbreaks, but I doubt that a similar pattern will be there in the next outbreak, therefore the predictive power is very low. And this is probably true, which is one of the main reasons why fighting Aedes borne diseases is so extremely difficult: all neigborhoods should be vigilant.
Editorial remarks
Page 2 line 58. Replace “Economic Strata” with “Socio-economic strata (SES)”, to introduce the correct term and the abbreviation for the remainder of the paper (only in the caption of Table 2 the abbreviation is explained) Addressed
Page 3, paragraph 2.2.: Deducting from caption of y-axis in figure 2 and figure 5, I assume that the “date of notification” is used as temporal indicator for cases. Please add that to the description of cases in paragraph 2.2. Would day of onset of symptoms not be a better temporal indicator? Addressed
Page 4: I am curious to the rational behind the choice of including distance of residents of water bodies and parks in the analyses of Aedes aegypti. Why not distance to major highway, or cemetery or illegal dump or abandoned houses or……. The reasoning in the discussion page 15 394-395 that in “parks as areas of vegetation that are more likely to collect pools of standing water” is unfounded. Addressed
Page 5 line 171 : I am curious to the rational behind the choice of including “percentage of females” in the analyses. On page 15 line 401-402 the authors state that it is explained before but I did not find it. Addressed
Page 5 line 178. Due to the first sentence of this paragraph (2.5.4) that “the dataset was too sparse and noisy to fit a spatio-temporal model the monthly counts were aggregated over the entire study period” the authors should consider to amend the title into “Spatial heterogeneity..” in stead of “spatio-temporal” . Although the period runs from 2014-2016 and figures present temporal aspects these are not statistically analyzed. Addressed
Page 6 line 200-208 could be omitted as it contains the same (non interpretable) summary information as table 1. Addressed
Page 6 line 212-213 I do not understand this sentence, please rephrase. Addressed
Page 6 line 217; Please write chikungunya without capital letter c. Addressed
Page 8:Legend of Figure 3. The color scheme is puzzling to me. Why not going from dark red (high-high) light red (low high) to light blue (high low) to dark blue (low low). Same holds for figure 6. Not addressed and that makes me wonder whether I actually interpreted the four categories correctly. I looked in the paper and the four categories are not anywhere explained. Please add a legend for the four categories (high-high to low-low).
Page 9 line 274: Typo in line 274 or in table 2 (1.75 vs 1.74) Addressed as there are complete new calculations. I have no idea what the back ground is from the different outcome. Also the original figure 4 (new fig 4)and 7 (new figure 6). I also only now recognized that the color scheme of these two figures are rather odd as the categories have extremely different intervals. I would like the authors to elaborate why the intervals are choosen as they are, especially for ZIKS as the incidence intervals; there are 7 categories all together to cover incidence rate from 0-7.8, of which the authors used 3 categories to cover only range 0.9-1.6 ???
Page 15 line 406-408.I like to remind the authors that the key breeding sites in area is largely context dependent. The authors state storage containers holding up to 92% of the pupa populations as a constant percentage while this was an outcome of a survey at the beginning of the century somewhere else in Columbia. Addressed
Page 15- line 413-415: Unclear sentence, please rephrase Addressed
Page 16: Line 457-458. Unclear what the authors mean when listing “differences in vector, invasion, breeding and disease transmission Addressed

Author Response
Reviewer # 1 Comments.
As a medical entomologist working in Public health institution on Aedes vector in Caribbean and Europe, I read the manuscript by Mc Hale et al. submitted to IJERPH with great interest. The paper tells a compelling story, challenging the common hypotheses on that Aedes borne infections are driven by social vulnerability. I highly appreciate the daunting task of the authors to bring all the data together, analyse and model it and draw conclusions from it. However, I have some issues that I would like to raise and I hope the authors will be keen to answer them
Scientific questions
Unfortunately the authors avoided to address the most curious result of their study “why the spatial clustering for CHIKV cases differs from those for ZIKV”. They mention it in page 15, line 390-391 but provide no explanation or hypothesis for this finding; it is not even discussed. While reading the paper I was eager to learn what the authors’ take on the issue was, but it was not addressed. The authors treat the two outbreaks completely independent while they both are transmitted by the same vector. Would there be a spatial difference in inhabitants between clinical outcome after infection between neighborhoods. Would there be a reason to assume that people with higher SES would report Zika but not chikungunya. Would there be a reason that areas with SES where more difficult to access during the Zika outbreak then the chikungunya outbreak. I largely miss the discussion on this issue. Only partially addressed. The authors added info p22 lines481-483, and lines493-494, p23 line 562 (the best one). The authors admit to test correlations with potential risk factors with the incidence data but still no insights whether it is driven by mosquito density or reporting bias. Still puzzling also for me how the strong difference between CHIKV and ZIKV occurred but I am happy they addressed here but the authors “forgot” to implement it in the conclusion (therefore please omit “we clearly showed that living in the high SES was a risk factor for ZIKV transmission”; you CAN state that they are important hotpots of case incidence, acknowledging that the high incidence is either real or biased. The fact that the latter is not implemented in the text in the conclusion I get the impression that the authors are more data crunchers then public health professionals who wants to know what is the factor /cause behind the data so that interventions can be applied correctly. It is still weird that the incidence of diseases that are transmitted by the same vector have different environmental drivers. I wished the authors would be a bit more impressed with this difference. A statistical comparison between the CHIKV and ZIKV hotspots would be an interesting exercise.
Response:
In response to the reviewer’s comment we have altered lines 517-518 (page 17) in the Conclusion section to: ‘We clearly showed that living in the high SES was a risk factor for ZIKV case incidence, though this may have been driven by biased case reporting.’
We share the reviewer’s frustration with the difference between these two outbreaks. In the paper, we clearly discuss the limitation in how we obtained infected cases, namely, clinical suspicion which is inherently prone to bias. In the discussion, we discuss the possibility that our findings are due to bias and we also discuss the possibility that there was a difference in mosquito distribution or control during the two outbreaks. Unfortunately, we cannot say definitively what the underlying etiologic mechanism is. However, in Colombia and throughout the Americas clinical suspicion will be used far more than laboratory confirmation to identify cases and respond to outbreaks. Therefore, we believe these findings are important contribution to the body of literature that describes the epidemiologic trends of these infections in the Americas. We do not think that high SES solely can explain the epidemiologic trend we demonstrate during the ZIKV infection. However, this finding has prompted further analysis that we are currently conducting, which are targeting individual related factors that may be hindered by the geographical level of the presented analysis.
On the other hand, we consider quite unfortunate and incorrect, the reviewer’s comments about us being ‘data crunchers’ rather than public health professionals. We understand the reviewer is not familiar with our research works on the incidence and control of Aedes aegypti-borne diseases elsewhere and in Barranquilla with members of the Public Health Authorities. Let us bring in here some examples of our work on: a) the development of new robust, rapid and sensitive larvae and pupae vector surveillance methods now used throughout Colombia and the world, b) intervention strategies, c) development of new and far more accurate full IgM and IgG ELISA titrations, d) development of a simple robust and inexpensive DENV tetravalent dot-blot assay, e) early acute phase definitive prognostic criteria for cases who subsequently will develop DHF/DSS to then avert its onset, f) isolation of all four DENV serotypes in the city and location of the ‘hot-spots’ for the concentration of vector control efforts and g) the identification of dual-gender sexual transmission of ZIKV and we have now pioneered the identification of genetic recombination from full genome determinations in different DENV serotypes in these cases imported into the south-western neighbourhoods that were identified as hotspots in this manuscript (near the cargo warehouses) (see reference 77 Falconar and Romero-Vivas 2012 J.Clin. Med. Res. 4: 33-44; manuscript in prep). The latter work clearly showed that Barranquilla is a ‘mixing bowl’ of DENV genetic types due to the importation of multiple strains which then cause epidemics caused by multiple strains of different DENV serotypes starting around the cargo warehouses. These are therefore unique findings and will allow the public health vector control teams (Note that we have been working with Pedro Arango-Padilla for many years and who is a co-author on this paper and is also the Head of Vector Control of the Public Health Authority in Barranquilla) to focus on these areas to control these imported epidemics.
Falconar AK, de Plata E, Romero-Vivas CM. 2006. Altered enzyme-linked immunosorbent assay immunoglobulin M (IgM)/IgG optical density ratios can correctly classify all primary or secondary dengue virus infections 1 day after the onset of symptoms, when all of the viruses can be isolated. Clin Vaccine Immunol. 2006 13: 1044-1051.
Romero-Vivas CM, Arango-Padilla P, Falconar AK. 2006. Pupal-productivity surveys to identify the key container habitats of Aedes aegypti (L.) in Barranquilla, the principal seaport of Colombia. Ann Trop Med Parasitol. 2006 100 Suppl 1:S87-S95.
Alexander N, Lenhart AE, Romero-Vivas CM, Barbazan P, Morrison AC, Barrera R, Arredondo-Jiménez JI, Focks DA. 2006. Sample sizes for identifying the key types of container occupied by dengue-vector pupae: the use of entropy in analyses of compositional data. Ann Trop Med Parasitol. 100 Suppl 1:S5-S16.
Romero-Vivas CM, Llinás H, Falconar AK. 2007. Three calibration factors, applied to a rapid sweeping method, can accurately estimate Aedes aegypti (Diptera: Culicidae) pupal numbers in large water-storage containers at all temperatures at which dengue virus transmission occurs.
J Med Entomol. 2007 Nov;44(6):930-7.
Falconar AK, Romero-Vivas CM. 2012. Simple Prognostic Criteria can Definitively Identify Patients who Develop Severe Versus Non-Severe Dengue Disease, or Have Other Febrile Illnesses. J Clin Med Res. 2012 4: 33-44.
Sim S, Jupatanakul N, Ramirez JL, Kang S, Romero-Vivas CM, Mohammed H, Dimopoulos G. 2013. Transcriptomic profiling of diverse Aedes aegypti strains reveals increased basal-level immune activation in dengue virus-refractory populations and identifies novel virus-vector molecular interactions. PLoS Negl Trop Dis. 7: e2295.
Towers S, Brauer F, Castillo-Chavez C, Falconar AKI, Mubayi A, Romero-Vivas CME. 2016. Estimate of the reproduction number of the 2015 Zika virus outbreak in Barranquilla, Colombia, and estimation of the relative role of sexual transmission. Epidemics. 17: 50-55.
Romero-Vivas CM, Llinás H, Falconar AK. 2015. The single water-surface sweep estimation method accurately estimates very low (n = 4) to low-moderate (n = 25-100) and high (n > 100) Aedes aegypti (Diptera: Culicidae) pupae numbers in large water containers up to 13 times faster than the exhaustive sweep and total count method and without any sediment contamination. Trop Med Int Health. 20: 326-33.
Although the results indicated the differences I am not satisfied with the rational behind it and I am uncomfortable with the authors statement in the first sentence of the conclusion (p16, line 469-470) that the study provide evidence based framework that the PH programmes can use to effectively target drivers of future epidemic. As a public health expert I still wonder how I could act based on the findings of the paper. Partially addressed. I am wondering whether not just random factors determine the introduction of a virus in a neighbourhood and that the vector density in all neighbourhoods is above threshold for efficient transmission and for that reason other possible random factors play a more important role. The authors found a pattern in two outbreaks, but I doubt that a similar pattern will be there in the next outbreak, therefore the predictive power is very low. And this is probably true, which is one of the main reasons why fighting Aedes borne diseases is so extremely difficult: all neigborhoods should be vigilant.
Response:
We would resist the temptation to explain these described trends to chance. First, we have carefully described and formulated our statistical modelling approach to reduce the possibility that any of our findings are due to chance. While our paper may fail to establish causality for which factors may have been driving the epidemics, we believe that the trends described may allude to underlying drivers yet to be discovered or fully described. That is what random effects stand for, accounting for other unknown effects or factors that have not been measured and that may have driven the spatial distribution and intensity of these outbreaks. In the case of ZIKV, our model suggests the occurrence of other neighbourhood related spatial effects (then not independent) that have not been considered in here (DIC value for locally spatial random effect lower than DIC for the model accounting for independent effects). These other spatial effects are to be targeted in further research works. Figure S7 of the supplementary file shows the persistent cluster after accounting for the fixed effect and local spatial random effects.
At the contrary, for CHIKV outbreak, the variation on incidence distribution appears to also be highly explained by unaccounted factors which would be neighbourhood-specific and non-spatially correlated (independent effect).
In the event of future epidemics, we hope that our findings are used to compare and contrast differences in epidemics to further assist future public health responses. Having said this, it is important to remark that the purpose of this work was not to forecast future scenarios, we do not state this throughout the manuscript, but describe and identify potential risk factors for these CHIKV and ZIKV outbreaks.
Editorial remarks
Page 2 line 58. Replace “Economic Strata” with “Socio-economic strata (SES)”, to introduce the correct term and the abbreviation for the remainder of the paper (only in the caption of Table 2 the abbreviation is explained) Addressed
Page 3, paragraph 2.2.: Deducting from caption of y-axis in figure 2 and figure 5, I assume that the “date of notification” is used as temporal indicator for cases. Please add that to the description of cases in paragraph 2.2. Would day of onset of symptoms not be a better temporal indicator? Addressed
Page 4: I am curious to the rational behind the choice of including distance of residents of water bodies and parks in the analyses of Aedes aegypti. Why not distance to major highway, or cemetery or illegal dump or abandoned houses or……. The reasoning in the discussion page 15 394-395 that in “parks as areas of vegetation that are more likely to collect pools of standing water” is unfounded. Addressed
Page 5 line 171 : I am curious to the rational behind the choice of including “percentage of females” in the analyses. On page 15 line 401-402 the authors state that it is explained before but I did not find it. Addressed
Page 5 line 178. Due to the first sentence of this paragraph (2.5.4) that “the dataset was too sparse and noisy to fit a spatio-temporal model the monthly counts were aggregated over the entire study period” the authors should consider to amend the title into “Spatial heterogeneity..” instead of “spatio-temporal” . Although the period runs from 2014-2016 and figures present temporal aspects these are not statistically analyzed. Addressed
Page 6 line 200-208 could be omitted as it contains the same (non interpretable) summary information as table 1. Addressed
Page 6 line 212-213 I do not understand this sentence, please rephrase. Addressed
Page 6 line 217; Please write chikungunya without capital letter c. Addressed
Page 8:Legend of Figure 3. The color scheme is puzzling to me. Why not going from dark red (high-high) light red (low high) to light blue (high low) to dark blue (low low). Same holds for figure 6. Not addressed and that makes me wonder whether I actually interpreted the four categories correctly. I looked in the paper and the four categories are not anywhere explained. Please add a legend for the four categories (high-high to low-low).
Response:
We have now added:
Figure 3. Differential monthly LISA clusters of CHIKV infected cases during the period of peak incidence (July 2014 – January 2015). High-High to Low-Low colour-coded neighbourhoods (see Materials and Methods 2.5.2) were significant, defined as p < 0.05. NS: not significant (grey colour).
Figure 5. Differential monthly LISA clusters of ZIKV cases during the period of peak incidence (October 2015 – April 2016). High-High to Low-Low colour-coded neighbourhoods (see Materials and Methods 2.5.2) were significant, defined as p < 0.05. NS: not significant (grey colour)
Response:
Further explanation of these categories was added to section 2.5.2 of the Materials and Methods as:
An area of high-high correlation (hotspot) indicated that the estimated incidence exceeded the neighbourhood average. An area of high-low correlation indicates that the central neighbourhood was negatively correlated with its neighbours, having a higher than average incidence, compared to a lower than average incidence in the neighbours. Alternatively, low-low correlation indicated that neighbourhoods were correlated together with lower than average incidence. But low-high indicated negative correlation with the central neighbourhood having lower than average incidence compared to a higher than average incidence in its neighbours. More details about the implementation of these statistics are provided in supplementary file (Text S1).
Page 9 line 274: Typo in line 274 or in table 2 (1.75 vs 1.74) Addressed as there are complete new calculations. I have no idea what the back ground is from the different outcome. Also the original figure 4 (new fig 4) and 7 (new figure 6). I also only now recognized that the color scheme of these two figures are rather odd as the categories have extremely different intervals. I would like the authors to elaborate why the intervals are choosen as they are, especially for ZIKS as the incidence intervals; there are 7 categories all together to cover incidence rate from 0-7.8, of which the authors used 3 categories to cover only range 0.9-1.6 ???
Response:
In response to another reviewer’s comments, we have added the variable ‘distance from major roads’ to our multivariate models and rerun the Bayesian conditional autoregressive analysis. Therefore, our calculations are completely different from the original submission.
As demonstrated in the figures that we have now placed in the supplemental section (Figures S8 and S9), the size of the CHIKV and ZIKV were quite different. As noted in our results, Section 3.2, there were a total of 1865 CHIKV cases. Comparatively, as noted in our results, Section 3.3, there were a total of 7029 ZIKV cases. In addition, the distribution of cases across the map of Barranquilla are quite different. For this reason, we had to choose different breaks in our SIR in order to demonstrate where the cases were clustering. If we were to use the same breaks for both outbreaks, we would not be able to see the clustering of case incidence as well in one of the outbreaks.
Page 15 line 406-408.I like to remind the authors that the key breeding sites in area is largely context dependent. The authors state storage containers holding up to 92% of the pupa populations as a constant percentage while this was an outcome of a survey at the beginning of the century somewhere else in Columbia. Addressed
Page 15- line 413-415: Unclear sentence, please rephrase Addressed
Page 16: Line 457-458. Unclear what the authors mean when listing “differences in vector, invasion, breeding and disease transmission Addressed

Reviewer 2 Report
The authors have addressed the comments from the reviewer. In light of this new version some comments follow.
· Lns. 153-154: can the authors please clarify if the Euclidean distance from the neighborhood to the water bodies is calculated from centroid to centroid? It is clear that it is from the centroid of the neighborhood, but not to the centroid of the water body.
· Ln. 214: what does it mean that the “SES level was equal to low”?
· Table 1, column 1: the entry “distance from major roads to ward centroids”, there are no wards referenced elsewhere. Do the authors mean neighborhoods?
· Several of the maps are missing scale bars and north arrows.
Author Response
Reviewer # 2 Comments
The authors have addressed the comments from the reviewer. In light of this new version some comments follow.
· Lns. 153-154: can the authors please clarify if the Euclidean distance from the neighborhood to the water bodies is calculated from centroid to centroid? It is clear that it is from the centroid of the neighborhood, but not to the centroid of the water body.
We thank reviewer’s comment on this. The straight-line distance (Euclidean distance) was calculated from “the centroid of every neighbourhood to the nearest pixel of water identified in the MNDWI” surface created from Landsat 8 data. We have amended this in the paper (page 4, line 154).
· Ln. 214: what does it mean that the “SES level was equal to low”?
Response:
This simply means that the regression coefficients were calculated in reference to the lowest SES level. We have now changed the text in page 6, line 218-220 to read:
‘All explanatory covariates were mean centred. In this way the intercept was interpretable as the average global SIR and the SES regression coefficients were calculated in reference to the lowest SES level.’
· Table 1, column 1: the entry “distance from major roads to ward centroids”, there are no wards referenced elsewhere. Do the authors mean neighborhoods?
The text has been amended to replace the word ‘ward’ by ‘neighbourhood.’
· Several of the maps are missing scale bars and north arrows.
We have now included scale bars and compass in all the figures.

Reviewer 3 Report
I see the authors have improved the manuscript. Now it can be read. Thank you.
Contrast and criticisms are useful in peer review journals.
I still see significant limitations in the study. There is no proper scientific evidence in the Multivariate models of ecological studies to assure infection. I did not find the area of each cluster. If is in the manuscript, please indicate it.
Additionally from your SES associations of CHIKV and ZIKV infection, authors know infection comes from a mosquito, not from the environment. There are also some natural barriers from mosquitoes that are not listed in the study. The discussion should mention these relations about distance, natural barriers and infections. This must be stated in the manuscript prior to publication and it is a very important limitation due to the apparently large clusters in the study.
Author Response
Reviewer # 3 Comments
I see the authors have improved the manuscript. Now it can be read. Thank you.
Contrast and criticisms are useful in peer review journals.
I still see significant limitations in the study. There is no proper scientific evidence in the Multivariate models of ecological studies to assure infection. I did not find the area of each cluster. If is in the manuscript, please indicate it.
Given the financial constraints in Colombia, the vast majority of our cases come from clinical suspicion, as discussed in the methods. For this reason, we refer to ‘infected cases’ as opposed to incidence of infection, which we are not able to show in this paper. We outline this in section 2.2 of the methods and discuss related limitations in the fourth paragraph of the discussion.
Since this is an ecological study, the neighbourhood is our unit of analysis. Because of this, we do not believe it would be sensible to add the area of our neighbourhoods together, as this would not add meaningful data to our analysis. The neighbourhood structure used in our study are taken from municipal officials. We believe that this unit may be useful to public health officials in targeting their limited resources. The area of each neighbourhood would not provide additional data since the location and name of the neighbourhood are sufficient for resource allocation.
Additionally from your SES associations of CHIKV and ZIKV infection, authors know infection comes from a mosquito, not from the environment. There are also some natural barriers from mosquitoes that are not listed in the study. The discussion should mention these relations about distance, natural barriers and infections. This must be stated in the manuscript prior to publication and it is a very important limitation due to the apparently large clusters in the study.
In response to this reviewer’s comment, Barranquilla is just above sea level and has no natural barriers which would affect vector movement, but it is located only on the west bank of the Magdalena River (the other side of the Magdalena River has poor non-urban areas located in another State). We have stated this in the discussion (page 17, line 504-509):
‘Although adult Aedes aegypti are known to fly up to 2km if required, they usually fly much shorter distances (e.g. 10-150 metres) to find blood meals and breeding sites [91] and there were no natural barriers (e.g. large lakes, parks, hills or valleys) in this urban study site since we did not study any of the (only poor non-urban) neighbourhoods located on the opposite (eastern) bank of the Magdalena River.’
Added reference
91: Sallam M.F.; Fizer C.; Pilant A.N.; Whung P-Y. Systematic review: land cover, meteorological, and socioeconomic determinants of Aedes mosquito habitat for risk mapping. 2017 Int. J. Environ. Public Health 14, 1230.
